# A Workflow for Missing Values Imputation of Untargeted Metabolomics Data

**DOI:** 10.3390/metabo10120486

**Published:** 2020-11-26

**Authors:** Tariq Faquih, Maarten van Smeden, Jiao Luo, Saskia le Cessie, Gabi Kastenmüller, Jan Krumsiek, Raymond Noordam, Diana van Heemst, Frits R. Rosendaal, Astrid van Hylckama Vlieg, Ko Willems van Dijk, Dennis O. Mook-Kanamori

**Affiliations:** 1Department of Clinical Epidemiology, Leiden University Medical Center, Postal Zone C7-P, PO Box 9600, 2300 RC Leiden, The Netherlands; T.O.Faquih@lumc.nl (T.F.); J.Luo@lumc.nl (J.L.); S.le_Cessie@lumc.nl (S.l.C.); F.R.Rosendaal@lumc.nl (F.R.R.); A.van_Hylckama_Vlieg@lumc.nl (A.v.H.V.); 2Julius Center for Health Sciences and Primary Care, University Medical Centre Utrecht, Utrecht University, 8, 3584 Utrecht, The Netherlands; M.vanSmeden@umcutrecht.nl; 3Department of Biomedical Data Sciences, Section Medical Statistics and Bioinformatics, Leiden University Medical Center, 2, 2333 Leiden, The Netherlands; 4Institute of Bioinformatics and Systems Biology, Helmholtz-Zentrum München, 85764 Neuherberg, Germany; g.kastenmueller@helmholtz-muenchen.de; 5Institute of Experimental Genetics, Genome Analysis Center, Helmholtz Zentrum München, 85764 Neuherberg, Germany; 6Department of Physiology, Institute for Computational Biomedicine, Englander Institute for Precision Medicine, Weill Cornell Medicine, New York, NY 10065, USA; jak2043@med.cornell.edu; 7Department of Internal Medicine, Section of Gerontology and Geriatrics, Leiden University Medical Center, 2333ZA Leiden, The Netherlands; R.Noordam@lumc.nl (R.N.); D.van_Heemst@lumc.nl (D.v.H.); 8Einthoven Laboratory for Experimental Vascular Medicine, Leiden University Medical Center, 2, 2333 Leiden, The Netherlands; K.Willems_van_Dijk@lumc.nl; 9Department of Internal Medicine, Division of Endocrinology, Leiden University Medical Center, 2, 2333 Leiden, The Netherlands; 10Department of Human Genetics, Leiden University Medical Center, 2, 2333 Leiden, The Netherlands; 11Department of Public Health and Primary Care, Leiden University Medical Center, 2, 233 Leiden, The Netherlands; 12Metabolon Inc., Morrisville, NC 27560, USA

**Keywords:** imputation, multiple imputation using chained equations, k-nearest neighbors, untargeted metabolomics, metabolon, simulation, workflow

## Abstract

Metabolomics studies have seen a steady growth due to the development and implementation of affordable and high-quality metabolomics platforms. In large metabolite panels, measurement values are frequently missing and, if neglected or sub-optimally imputed, can cause biased study results. We provided a publicly available, user-friendly *R* script to streamline the imputation of missing endogenous, unannotated, and xenobiotic metabolites. We evaluated the multivariate imputation by chained equations (MICE) and k-nearest neighbors (kNN) analyses implemented in our script by simulations using measured metabolites data from the Netherlands Epidemiology of Obesity (NEO) study (*n* = 599). We simulated missing values in four unique metabolites from different pathways with different correlation structures in three sample sizes (599, 150, 50) with three missing percentages (15%, 30%, 60%), and using two missing mechanisms (completely at random and not at random). Based on the simulations, we found that for MICE, larger sample size was the primary factor decreasing bias and error. For kNN, the primary factor reducing bias and error was the metabolite correlation with its predictor metabolites. MICE provided consistently higher performance measures particularly for larger datasets (*n* > 50). In conclusion, we presented an imputation workflow in a publicly available *R* script to impute untargeted metabolomics data. Our simulations provided insight into the effects of sample size, percentage missing, and correlation structure on the accuracy of the two imputation methods.

## 1. Introduction

Metabolomics studies have seen a steady growth due to the development and implementation of affordable and high-quality metabolomics platforms. These platforms can be split into two categories: targeted and untargeted metabolomics platforms based on their approach to metabolite identification [1,2]. Targeted platforms are focused on a known prespecified set of metabolites, while untargeted platforms aim to detect as many metabolites as possible in the sample without the need for explicit prior knowledge of their identity. The metabolite signatures detected (i.e., mass to charge ratio, m/z, or retention times) are subsequently matched in a metabolite library to determine their identity. Currently, both targeted and untargeted platforms can detect over 1000 metabolites in a single biological sample (e.g., blood, saliva, and urine). A typical issue with both these platform types is missing values from the measurement.

Missing values in metabolomics data are problematic for subsequent analyses, may be neglected, and are often mishandled or ignored. A common misconception is that missing values in metabolomics data are exclusively due to metabolites with a very low concentration, i.e., below the limit of detection of the instrument. Although in many circumstances the majority of missing values can be due to low concentrations, it has been shown that missing values can also be caused by biological and/or technical variation [3,4,5]. Based on the assumption that not reaching the limit of detection exclusively causes missingness, missing values are often handled with one or more of the following procedures:(1)For each metabolite the missing values are replaced (“imputed”) with a single value, such as the minimum detection level or half the minimum detection level. This approach results in overrepresentation of a single value in the population distribution. This may affect subsequent analyses and may cause biased results, regardless of the cause of missing values [5,6]. Furthermore, metabolites could be missing in some individuals because they are not biologically present in their system. Therefore, imputing these missing values will cause bias in the analysis. For example, if the metabolites for metformin are imputed, both diabetic patients who use the drug as well as and non-diabetic individuals who do not use the drug will have values for the it. This is a prominent issue in platforms such as Metabolon™ (Metabolon Inc., Durham, NC, USA) that include xenobiotic metabolites (e.g., metabolites from external sources such as medications).(2)Metabolites with a missing percentage above an arbitrary cut-off value (for example 20%) are removed from the dataset due to “too much missingness” regardless of the metabolite identity. By applying a cut-off above which metabolites are removed from the dataset, or, in the most extreme case only using the complete cases, data are unnecessarily discarded, that could have been of importance to the research question. Furthermore, this exclusion can affect further pathway analysis, such as metabolite set enrichment analysis, that explore possible pathway connections for the measured metabolites [7].

Several studies have evaluated imputation methods for metabolomics data. The consensus from these studies has so far been that imputation using half the minimum value leads to more bias than other methods and, consequently, this method is discouraged [3,8]. One alternative imputation method that has been recommended for metabolomics is the k-nearest neighbors (kNN) imputation [6,9]. An extensive simulation was performed that evaluated and compared 31 methods of imputation in a simulated untargeted metabolomics data provided by the Metabolon™ platform [6]. These methods included univariate methods such as half-minimum imputation and multivariate methods such as variations of kNN and multivariate imputation by chained equations (MICE). Two methods were concluded to have the best performance:(1)kNN on observations with variable pre-selection (“kNN-obs-sel”), a two-step method that incorporates the standard kNN algorithm with a preselection of a group of metabolites that are most correlated with the metabolite with missing values (i.e., auxiliary metabolites). Therefore, the neighbors selected by kNN will have similar metabolomic profiles [6].(2)MICE using the predictive mean matching method (“MICE-pmm”). Like kNN-obs-sel, the most correlated metabolites were used for the imputation. The imputed values are then selected from distribution of possible values to produce multiple imputed datasets [10,11,12].

In this paper, we expand upon the meticulous evaluation of the imputation methods by Do et al. [6], which was performed on an older version of the metabolomics platform that detects a smaller set of metabolites (*n* = 517). Furthermore, we set out to take unannotated (i.e., unidentified metabolites in the library) and xenobiotic metabolites into account. The recent Metabolon™ panel in use (Discovery HD4) has increased the number of metabolites to >1000, which includes more unannotated and xenobiotic metabolites. As more scientists are using metabolomics data in their research, it is helpful to have a user-friendly workflow for imputation using the best available methods. We provided this imputation workflow and a user-friendly *R* script to streamline the imputation of the Metabolon™ HD4 panel using kNN-obs-sel and MICE-pmm. Furthermore, we evaluated the imputations by the script in several scenarios with different missingness conditions by a resampling simulation analysis using measured metabolomics data from the Netherlands Epidemiology of Obesity (NEO) study.

## 2. Results

### 2.1. Metabolomic Data Characteristics

Metabolomics measurements in 599 individuals between the ages of 45 and 65 with normal BMI distribution from the NEO study identified 1365 metabolites. Detailed information regarding the population are provided in the Methods section and Appendix A. Known metabolites were annotated with their chemical name, super pathway, sub pathway, compound identifiers from various metabolite databases, and information regarding their biochemical properties. A total of 840 metabolites were from various endogenous pathways, 229 metabolites were characterized as xenobiotics, and 296 metabolites were unannotated (lacking information regarding chemical name and pathway). Of the 1365 identified metabolites, 800 (58.6%) contained missing values and the median number of missing metabolites per observation was 228 (38%) (Table 1).

We plotted the distribution of missing values in each metabolite group (Figure 1). The distribution of the number of missing values of the unannotated metabolites was similar to that of the endogenous metabolites rather than the xenobiotic metabolites. This suggests that most unannotated metabolites are most likely from an endogenous source, similar to the annotated endogenous metabolites, and are most likely expected to be present in all our participants.

The Pearson’s pairwise-complete correlation matrix for the endogenous and unannotated metabolites was calculated using all the metabolites (complete with no missing values and incomplete). For each incomplete metabolite we selected up to 10 complete metabolites with the highest absolute Pearson’s correlation (auxiliary metabolites). If the metabolite was not correlated with 10 metabolites (due to high missingness), then we selected the available correlated metabolites. We then calculated the mean value of the Pearson’s correlations for these metabolites. Figure 2 shows the distribution of the mean of the auxiliary absolute correlations with further details in Table A1. The 82% of the incomplete metabolites had a mean absolute Pearson’s correlation coefficient lower than 0.5 with their auxiliary metabolites. Overall, the median of the median absolute Pearson’s correlation coefficient was 0.4 (0.09–0.89), indicating a generally low intercorrelation between the metabolites.

### 2.2. Availability

The imputation script [13] streamlines the workflow by calculating the correlation matrix, selecting the auxiliary metabolites, and imputing the missing values of the metabolites using the provided data from the user. The script requires a dataset, a list of xenobiotic and non-xenobiotic metabolites (endogenous/unannotated), and a choice for the method of imputation (MICE-pmm or kNN-obs-sel). The script and example files can be found at: https://github.com/tofaquih/imputation_of_untargeted_metabolites.

### 2.3. Performance Evaluation

To evaluate our imputation framework, we applied it to impute metabolites with missing values in the measured NEO dataset (*n* = 599) using kNN-obs-sel and MICE-pmm. All metabolites were imputed apart from 12 metabolites (3 endogenous, 9 unannotated) in the dataset that had >90% missingness and were subsequently treated as xenobiotic and imputed to 0. As mentioned in the Methods section, extremely high missingness limits the amount of data needed to impute the metabolites and to find auxiliary metabolites. High missingness in the 3 endogenous metabolites could have been caused by technical or biological issues, or they could represent misannotated xenobiotic metabolites. The 9 unannotated metabolites were likely xenobiotic metabolites.

Simulations were performed to compare the performance of the imputation method (MICE-pmm or kNN-obs-sel). As detailed in the Methods section, we generated 144 resampling simulation scenarios, using four metabolites from independent pathways and varying mean correlations with auxiliary metabolites (PC(32:2) (mean absolute correlation = 0.64), urate (mean absolute correlation = 0.49), glutamate (mean absolute correlation = 0.49), succinylcarnitine (mean absolute correlation = 0.36)), three sample sizes (50, 150, 599), three percentage of missing (15%, 30%, 60%), and two missing mechanisms missing mechanisms (missing completely at random (MCAR) and probabilistic limit of detection (PLoD)). The percentage biases from the simulation are presented in Figure 3 and Table 2. Root mean squared errors (RMSE) are shown in Figure 4, Table A5, and Table A6. The mean and standard deviation of the estimates from the simulation are provided in Table A3 and Table A4 using MCAR and PLoD mechanisms, respectively. We used nested loop plots [14] to produce all the figures.

#### 2.3.1. Sample Size

We observed a decrease of bias and RMSE as the sample size increased (Figure 3 and Figure 4). This trend was consistent for MICE-pmm for each metabolite, with the percentage bias median (range): 8.2 (0.3–87.4) in *n* = 50 decreasing to median (range): 0.8 (0.1–9.7) in *n* = 599. However, increasing sample size did not improve imputation with kNN-obs-sel. Overall, percentage bias was median (range): 4.7 (0.2–66.1) in *n* = 50, median (range): 5.7 (0.5–72.7) in *n* = 150, and median (range): 5.2 (0.1–54.3). Furthermore, in some scenarios, bias and RMSE increased in larger sample sizes even with the same missing percentage and missing mechanism; this was particularly noticeable for succinylcarnitine (mean absolute correlation = 0.36) where the percentage bias increased in 60% missing from 8.3% in *n* = 50, to 46.1% in *n* = 150, and 54.3% in *n* = 599. Finally, for sample sizes of *n* = 50 and 150, MICE-pmm had lower bias than kNN-obs-sel but a RMSE higher or similar to kNN-obs-sel.

#### 2.3.2. Percentage of Missing

In scenarios with 15% and 30% missing, MICE-pmm and kNN-obs-sel showed low bias and RMSE across all sample sizes. At 15% missing, MICE-pmm had a percentage bias of median (range): 0.7 (0.1–12.7), while kNN-obs-sel had a percentage bias of median (range): 1.9 (0.1–13.8). At 30% missing, MICE-pmm had a percentage bias of median (range): 2.0 (0.1–26.2) and kNN-obs-sel had a percentage bias of median (range): 4.4 (1.1–31.9). Finally, in 60% missing MICE-pmm had a percentage bias of median (range): 7.8 (0.1–87.4) and kNN-obs-sel had a percentage bias of median (range): 14.7 (1.9–72.7). Overall, MICE-pmm had lower bias in all missing percentages than kNN-obs-sel. However, the percentage bias for kNN-obs-sel was often lower than that of MICE-pmm at 30% and 60% missing in *n* = 50.

#### 2.3.3. Correlation Strength with the Auxiliary Metabolites

We compared the percentage bias and RMSE of both imputation methods for the four metabolites to assess the influence of correlation strength of the auxiliary metabolites as shown in Table 2 and Table A2 and Figure 3 and Figure 4. We observed that availability of auxiliary metabolites with higher correlation for the imputation greatly reduced the bias and RMSE in both methods. In PC(32:2), the metabolite with the highest mean correlation (mean absolute correlation = 0.64), had the lowest bias overall. Percentage bias was median (range): 1.9 (0.2–19.5) with the MICE-pmm imputation and median (range): 2.3 (0.1–14.5) with kNN-obs-sel imputation. Glutamate (mean absolute correlation = 0.49) had median (range): 2.1 (0.1–40.2) percentage bias with MICE-pmm imputation and median (range): 2.7 (0.2–15.1) with kNN-obs-sel. Similarly, imputation of urate (mean absolute correlation = 0.49) using MICE-pmm had median (range): 3.8 (0.1–87.4) percentage bias and median (range): 6.2 (0.5–66.1) using kNN-obs-sel. In contrast, the percentage bias was much higher for the metabolite with the lowest mean correlation, Succinylcarnitine (mean absolute correlation = 0.36), with median (range): 2.6 (0.1–34.8) percentage bias using MICE-pmm imputation and median (range): 15.5 (1.8–72.7) with kNN-obs-sel. Moreover, the bias reached very high percentages in urate and succinylcarnitine compared to PC(32:2) and glutamate in the *n* = 50 subset.

#### 2.3.4. Missing Mechanisms

We used two mechanisms for missingness, MCAR and PLoD, in our simulations. Since PLoD is fundamentally missing not at random (MNAR), causing lower concentrations to have a higher likelihood of missingness, we examined how PLoD affects the performance of MICE-pmm and kNN-obs-sel compared to MCAR scenarios. MCAR scenarios had a percentage bias median (range): 1.9 (0.1–40.2) with MICE-pmm imputation and median (range): 5.3 (0.2–72.7) with kNN-obs-sel. PLoD scenarios had a percentage bias median (range): 3.2 (~0–87.4) with MICE-pmm imputation and median (range): 4.3 (0.1–66.1) with kNN-obs-sel. However, the RMSE (Figure 4, Table A5 and Table A6) was lower in PLoD for MICE-pmm (median (range): 11.1 (1.8–79)) than in MCAR (median (range): 14.3 (2.0–78.0)) and similarly lower for kNN-obs-sel in PLoD scenarios (median (range): 12.1 (2.1–70.3)) than MCAR (median (range): 16.5 (2.6–103.9)). Overall, imputing in PLoD scenarios lead to higher bias but lower RMSE compared to MCAR.

## 3. Discussion

Several simulation studies have evaluated different imputation methods for missing data in metabolomic datasets [3,6,9,15,16]. Nevertheless, the “half the minimum” method of imputation remains in use despite studies showing its sub-optimal performance [3,6,9,15,16]. In this study, we followed up on previous work and provided a framework and complementary *R* script on GitHub [13] that streamlines the imputation of untargeted metabolomics data. The script provides univariate imputation of zero for missing values considered to be truly absent in xenobiotics and two options of multivariate imputation methods for the remaining metabolites.

Overall, for the four metabolites we used in the simulation, we observed several factors that influenced the performance of each imputation method with different degrees. In the four metabolites we used, MICE-pmm performed better overall across different simulated scenarios. This performance is especially better in PLoD, which represents a missing mechanism similar to that of real metabolomics data [6]. MICE-pmm performance decreased the most in smaller sample sizes, somewhat less by the metabolite auxiliary correlation and the least by the missing percentage. Interestingly, the negative effect of missing percentage diminished as the sample size increased (*n* = 150 and *n* = 599). On the other hand, unlike MICE-pmm, kNN-obs-sel performance was decreased most by a higher percentage of missingness and low metabolite auxiliary correlation, which was not improved by increased sample size. A possible explanation is the nature of the kNN-obs-sel method. kNN-obs-sel focused on finding the nearest neighbors based on the correlated metabolites. If it failed to find strongly correlated metabolites, due to the metabolite naturally a having low correlation or due to a large amount of missing values, it selected weak neighbors. Therefore, even at larger sample sizes (150 and 599) the performance of the kNN-obs-sel method remained poor if the missing percentage was large and the metabolite had poor correlation.

### 3.1. Advantages and Disadvantages of MICE-pmm for Metabolomics

Unlike kNN imputations, we found few papers in the literature regarding the use of MICE imputation for metabolomics. The MICE-pmm imputation is a more intricate method for generating the imputation values. First, the imputation is repeated multiple times in order to assess the uncertainty of the imputation and provide standard errors of the estimates. Second, MICE-pmm imputation is more compatible with both normally distributed and skewed metabolites than kNN [10]. Third, MICE imputation utilizes discrete and continues variables for imputation. Therefore, MICE-pmm can include additional biologically relevant predictors and the outcome of the analysis of interest, improving the quality of the imputation [17]. These features explain the robustness of MICE-pmm in situations with low correlated auxiliary variables and high missingness.

However, MICE-pmm has some disadvantages. First, small sample sizes negatively affected the performance of MICE-pmm because this forces duplication and reuse of the same individuals [10]. Second, MICE imputation may require more computational run time and is somewhat more complicated to use than kNN because multiple imputed datasets are generated that require a pooling step for the analysis. We shortened computational time by using the latest *MICE R* package and by setting the number of multiple imputations to 5, which has been shown to be a suitable number of imputations [10]. This caused the running time for the complete imputation using MICE-pmm to be equal to that of kNN-obs-sel for the NEO dataset (*n* = 599). Furthermore, to test the speed of the script, we duplicated and stacked the NEO dataset to create larger datasets (*n* = 5400 and *n* = 20,000); MICE-pmm completed the imputations faster than kNN-obs-sel (Table A7). Third, with MICE-pmm it is not possible to apply further analysis such as lasso regression or random forest, which are common analysis methods used in metabolomics [7,18,19]. This is because MICE-pmm uses multiple datasets with Rubin’s Rules to pool the estimates of the analysis per dataset. One solution is to use the kNN-obs-sel method, as it always creates a single dataset for analysis. A second alternative would be to use MICE-pmm with a single imputation [m = 1], which can be specified in our script, and use that single dataset in the multivariate analysis. It should be noted that MICE-pmm with m = 1 still performed better than kNN-obs-sel for the larger sample sizes (see Table A2 and Table A6 and Figure A1 and Figure A2).

### 3.2. Limitations

Several methodological issues should be considered. Firstly, our evaluation was done using 599 samples, limited by available metabolomics data in the NEO study. Although this number is not particularly small, future research should be performed in larger datasets. Secondly, we assumed that all missing xenobiotics values are truly missing and replaced them by zero. This could be explored further by incorporating MICE-pmm or kNN-obs-sel to specifically impute xenobiotic metabolites from the same medication sources in persons taking the medication. Furthermore, it could be possible to use questionnaire and clinical data as imputation predictors in MICE-pmm to impute related xenobiotic metabolites. Thirdly, we did not explore alternative methods for MICE to handle small data sizes, such as regulation and penalization. Fourth, our simulation did not evaluate the variance estimators such as type-I and type-II errors or confidence interval coverage. Fifth, metabolites with very large missingness will have high bias and error in the imputation and should be interpreted with caution. Finally, the data do not provide the explicit cause of the missing values and, therefore, we could only assume if the values were truly missing, missing completely at random, or missing due to other reasons. Future studies which explore the causes of missingness will also allow us to impute the missingness more effectively.

## 4. Materials and Methods

### 4.1. Population Characteristics

The resampling simulation analyses were performed in the NEO study. This study has been extensively described elsewhere [20] and in Appendix A. The NEO study was accepted by the Medical Ethics committee of the Leiden University Medical Center under protocol P08.109. The study is also registered at clinicaltrials.gov under number NL21981.058.08/P08.109. All participants gave written informed consent [20]. Fasting state serum samples from a sub-population (*n* = 599) of the NEO study were sent for untargeted metabolomics measurements at Metabolon Inc. (Durham, NC, USA) using their Metabolon™ Discovery HD4 platform. In brief, this process involves four independent ultra-high-performance liquid chromatography mass spectrometry (UHPLC-MS/MS) platforms [21,22]. Two platforms used positive ionization reverse phase chromatography, one used negative ionization reverse phase chromatography, and one used hydrophilic interaction liquid chromatography (HILIC) negative ionization [22]. In total, 1365 serum metabolites were measured which included 840 endogenous, 296 unannotated, and 229 xenobiotic metabolites.

### 4.2. Imputation Methods

Following our examination of the missing data distribution in the NEO study (Figure 2), we decided the xenobiotic metabolites and non-xenobiotic metabolites (endogenous/unannotated) with different imputations. For xenobiotic metabolites, we assumed missing values are truly missing values. For example, when a medication metabolite concentration is missing, it is most likely that the participant is not taking the medication. Therefore, we decided to impute xenobiotic metabolites to zero, as imputing the values (with MICE, kNN, or half-min) would cause bias due to skewed distribution and false positives. For the non-xenobiotic metabolites (endogenous/unannotated), the missing pattern suggests that the unannotated metabolites are most likely endogenous. Therefore, we decided to impute the endogenous and the unannotated metabolites as a single group using the multivariate imputation methods of MICE-pmm and kNN-obs-sel. For these two multivariate methods, we first estimated a correlation matrix for all applicable/non-xenobiotic metabolites from which to select 10 auxiliary metabolites to be used for imputation.

For non-xenobiotic metabolites, we assumed that they are metabolites with truly missing values only if less than 90% of values were missing. This cut-off was necessary for multiple reasons: (1) it became nearly impossible to find auxiliary metabolites for imputation, (2) unannotated metabolites with high missing values are likely xenobiotic and therefore most likely truly missing, and (3) it became statistically problematic to perform multivariate imputation with such high missingness—particularly in small sample sizes [23].

In this study, we used MICE-pmm with 10 auxiliary metabolites to impute the missing values and generated 5 imputed datasets (m = 5). In addition to the auxiliary metabolites, we included further predictors by adding the clinical variables for the outcome (BMI) and the covariates (age and sex) used in the analysis model for the MICE-pmm imputation. The addition of these variables is required in MICE imputations to avoid bias in the results [23,24]. We used kNN-obs-sel only with 10 auxiliary metabolites to impute the missing values. Details regarding the imputation methods are provided in Appendix A. In our script, we incorporated the *R* package *mice* version 3.6.0 [10] for the MICE-pmm imputations and the package *VIM* version 4.8.0 [25] in the kNN-obs-sel imputations.

### 4.3. Evaluation Analysis and Missing Value Simulation

For the simulation, the analysis of interest was an ordinary least squares regression model with body mass index (BMI) as the outcome and age, sex, and a selected metabolite as the exposures. For the purpose of our study, BMI was used as the outcome for two reasons: (1) BMI is a variable that was measured in all our participants, and (2) BMI is strongly associated with many metabolites and commonly studied in metabolomics [26].

Four metabolites were used, selected based on the following criteria: (1) the metabolite had no missing values in the original NEO dataset, (2) the metabolite must have a strong association with BMI in our Metabolon™ data as well as in the literature using Metabolon™ [26], (3) the four metabolites must be from different biological pathways, and (4) the metabolites must have different mean correlations with their auxiliary metabolites. We found 6 out of 473 complete endogenous metabolites in NEO that fulfilled these criteria. We then narrowed the selection to one metabolite per pathway. Accordingly, we selected four metabolites: PC(32:2) (mean absolute correlation 0.64) from the lipid super pathway; succinylcarnitine (mean absolute correlation = 0.36) from the energy super pathway, the nucleotide urate (mean absolute correlation 0.49), and the amino acid glutamate (mean absolute correlation 0.49). Information regarding the metabolites is provided in Table 3.

We compared the performance of the two imputation methods by simulating missing values using the NEO dataset (*n* = 599). All simulations were performed on three datasets: the original dataset of 599 participants, and on two randomly sampled sub datasets of size *n* = 150 and *n* = 50. The distribution of age, sex, and BMI was maintained in the sub datasets of 50 and 150 individuals. We used the same sub datasets for the all corresponding simulation scenarios. Generating the subsets with different random sampling did not change the estimates drastically (not shown). It should be pointed out that the selected auxiliary metabolites differed slightly between the sub datasets. Metabolite levels were log transformed and standardized (mean of 0 and variance of 1). We calculated the estimates for each metabolite in the complete datasets separately to be used later for the bias and RMSE calculations. In the different simulation scenarios, we induced different percentages of missingness (15, 30, and 60%), and under two different mechanisms, MCAR and PLoD. In the PLoD missing scenarios, the odds of a value being missing increased as the concentration decreases. The total number of missing values was divided per quantile of the metabolite as follows: 40% into the lower quantile, 50% into the middle quantile, and 10% in the upper quantile.

The evaluation was done by (1) performing the linear regression analysis and obtaining the estimate of the regression coefficient using the complete metabolites data in each subset (Table 3), (2) simulating missing values, (3) imputing missing values using the two imputation methods, (4) estimating the regression coefficient using the imputed data, and (5) evaluating the difference between the estimate of the complete data for that subset and the estimate using the imputed methods, (6) repeating step 2 to 5 1000 times per simulation scenario. The performance of the imputation methods was evaluated using the following measures: raw bias, which is the difference between the real estimate and the mean of the simulations estimates, which can be a positive or a negative value; percentage bias, which is the raw bias divided by real estimate for easier interpretation and comparison [27]; the RMSE, which is the square root of the mean squared difference between estimated; and true value, this measure combines the bias and variance of the simulated estimates into a single measure and represents the precision of the method [28] (Appendix A).

Thus, in total, we used three datasets (*n* = 50, 150, 599), four metabolites (PC(32:2), succinylcarnitine, urate, glutamate), three missingness percentages (15%, 30%, 60%), two missing mechanisms (MCAR and PLoD), and evaluation by two imputation methods (kNN-obs-sel and MICE-pmm) for a total of 144 possible scenarios. Each of these scenarios was repeated 1000 times.

### 4.4. Imputation Workflow

To simplify the procedure of imputing missing data, we wrote an *R* script that calculates the correlation matrix between the different metabolites, selects the auxiliary metabolites with the largest correlation, imputes the xenobiotic metabolites with univariate imputation, and imputes the endogenous metabolites with a multivariate imputation (either kNN-obs-sel or MICE-pmm), which can be found on our GitHub repository [13].

## 5. Conclusions

In conclusion, we provided a workflow for handling missing values in untargeted metabolomics data using univariate imputation for xenobiotics and multivariate imputation using MICE-pmm or kNN-obs-sel for endogenous and unannotated metabolites. We further evaluated MICE-pmm and kNN-obs-sel in different simulated scenarios. Our evaluation showed that the performance of both methods is affected by three different factors, namely the metabolite mean correlation with auxiliary metabolites, the sample size, and the missing percentage. For MICE-pmm, sample size was the primary factor affecting bias and error inversely. For kNN-obs-sel, the primary factor affecting bias and RMSE was the metabolite correlation with the predictors, which, when high, can provide low bias and RMSE even in small sample sizes (*n* = 50). Since most of our metabolites had low mean correlation, MICE-pmm provided consistently higher performance measures than kNN-obs-sel and, as a result, we suggest using MICE-pmm imputation for untargeted metabolomics, particularly for larger datasets (*n* > 50).

## Figures and Tables

**Figure 1 metabolites-10-00486-f001:**
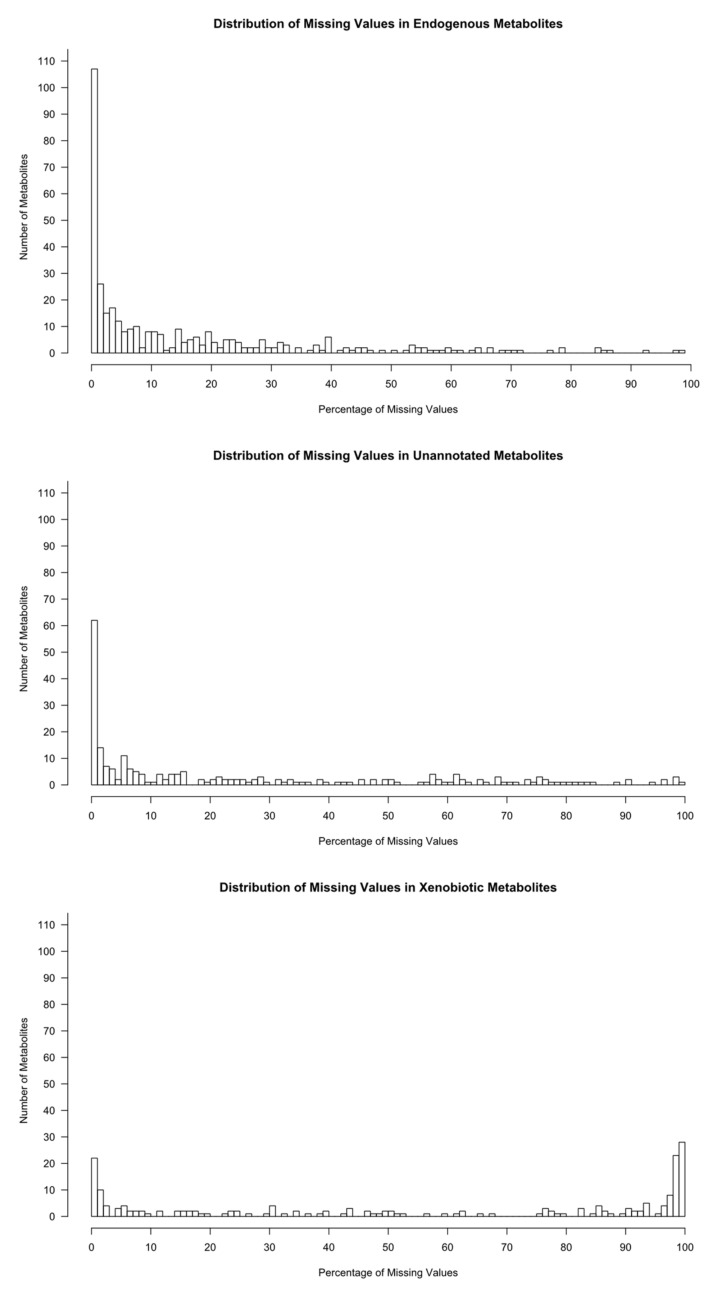
Distribution of the missing values in each metabolite group.

**Figure 2 metabolites-10-00486-f002:**
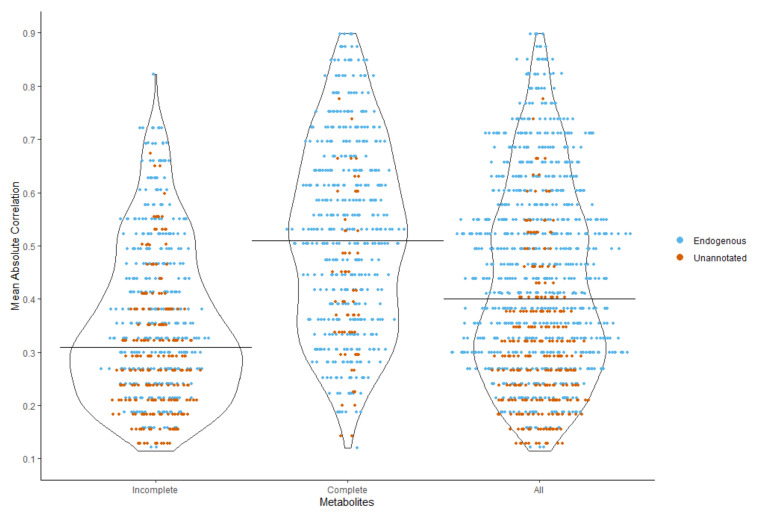
Distribution of the mean absolute correlations for the complete (without missing values) and incomplete (with missing values) endogenous and unannotated metabolites in the NEO dataset.

**Figure 3 metabolites-10-00486-f003:**
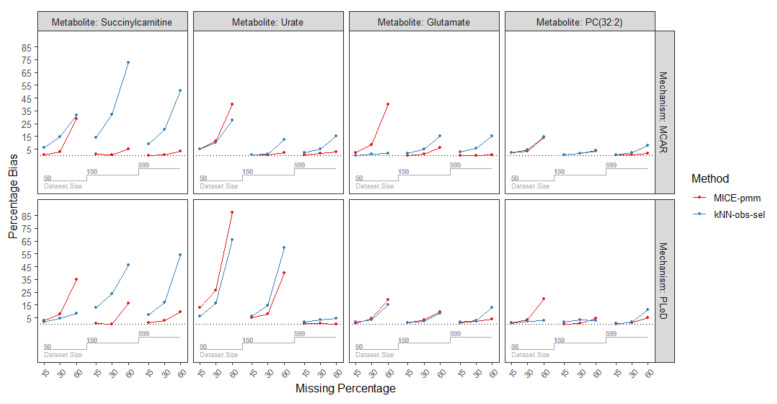
Nested loop plot of the percentage bias of the four metabolites from the simulation. The horizontal axis in each box represents the missing percentage and is split per sample size. Abbreviations: MCAR: missing completely at random; PLoD: probabilistic limit of detection.

**Figure 4 metabolites-10-00486-f004:**
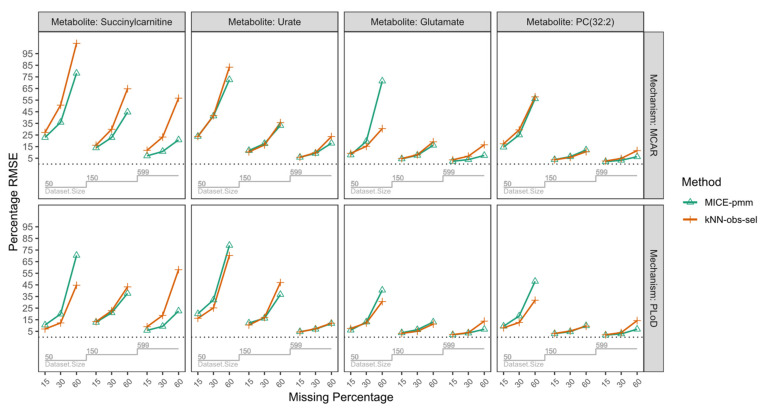
Nested loop plot of the root mean squared error (RMSE) of the four metabolites from the simulation. To simplify comparability in the plot we converted the RMSE values to a percentage by subtracting then dividing the RMSE values by the corresponding true estimates (in sample size *n* = 599). The horizonal axis in each box represents the missing percentage and is split per sample size. Abbreviations: MCAR: missing completely at random; PLoD: probabilistic limit of detection.

**Table 1 metabolites-10-00486-t001:** Summary of missing data in the Netherlands Epidemiology of Obesity (NEO) study.

Missing Data	Metabolite Groups			
	Endogenous (*n* = 840)	Unannotated (*n* = 296)	Xenobiotics (*n* = 229)	Total (*n* = 1365)
Metabolites with missing values, *n* (%)	367 (43.7)	236 (79.7)	197 (86.0)	800 (58.6)
Missing metabolites per observation, median (range)	57 (23–94)	59 (31–112)	110 (79–149)	228 (152–343)

In the NEO study, 1365 metabolites were measured in 599 individuals (observations).

**Table 2 metabolites-10-00486-t002:** Percentage bias of the imputation methods across different parameters on different metabolites including multivariate imputation by chained equations (MICE)-pmm with a single imputation.

Missing Mechanism	Sample Size	Missing Percentage	Metabolites/Imputation Method							
			PC(32:2)		Succinylcarnitine		Glutamate		Urate	
			MICE-pmm	kNN-obs-sel	MICE-pmm	kNN-obs-sel	MICE-pmm	kNN-obs-sel	MICE-pmm	kNN-obs-sel
MCAR	*n* = 50	15%	2.0	2.1	0.3	6.0	2.2	0.2	4.9	4.9
		30%	3.2	4.5	2.8	14.4	8.6	1.1	11.3	10.2
		60%	13.9	14.5	28.9	31.3	40.2	2.0	39.9	27.6
	*n* = 150	15%	0.7	0.5	1.4	13.8	0.2	1.9	0.7	0.5
		30%	1.9	1.8	0.4	31.9	1.1	5.1	0.7	1.3
		60%	3.9	3.2	5.2	72.7	6.2	15.0	2.09	12.5
	*n* = 599	15%	0.3	0.8	0.1	9.3	0.2	2.6	0.6	2.2
		30%	0.7	2.4	0.6	20.1	0.1	5.6	1.5	4.9
		60%	1.9	7.9	3.5	50.9	0.7	15.1	2.6	15.0
PLoD	*n* = 50	15%	1.3	0.8	2.8	1.8	0.7	1.8	12.7	6.2
		30%	3.5	2.2	7.7	4.3	4.2	3.6	26.3	16.5
		60%	19.5	3.0	34.8	8.3	19.3	15.0	87.4	66.1
	*n* = 150	15%	0.2	1.6	0.7	12.8	1.3	0.8	5.3	6.3
		30%	0.3	3.3	0.2	23.4	3.0	2.3	8.0	14.9
		60%	4.2	2.6	16.2	46.1	9.4	8.6	39.8	59.7
	*n* = 599	15%	0.5	0.1	0.9	7.3	0.9	1.4	0.5	1.9
		30%	1.1	1.6	2.5	16.7	2.1	2.9	0.6	3.3
		60%	4.9	11.5	9.7	54.3	4.1	13.1	0.1	4.3

Abbreviations: MCAR: missing completely at random; PLoD: probabilistic limit of detection.

**Table 3 metabolites-10-00486-t003:** Properties of the selected metabolites for the simulation.

Metabolite Full Name	Mean Absolute Correlation	Super Pathway	Sub Pathway	Estimate *n* = 599	Estimate *n* = 150	Estimate *n* = 50
PC(32:2)	0.64	Lipid	Plasmalogen	−4.18 × 10^−7^	−3.64 × 10^−7^	−4.38 × 10^−7^
Urate	0.49	Nucleotide	Purine Metabolism	1.39 × 10^−8^	9.58 × 10^−9^	9.69 × 10^−9^
Glutamate	0.49	Amino Acid	Glutamate Metabolism	1.83 × 10^−7^	2.89 × 10^−8^	1.66 × 10^−8^
Succinylcarnitine	0.36	Energy	TCA Cycle	2.84 × 10^−6^	1.53 × 10^−6^	4.53 × 10^−6^

Abbreviations. Mean absolute correlation: mean of the 10 absolute Pearson’s correlations from the metabolite correlation matrix. Estimate is the regression coefficient from the model BMI~age + sex + metabolite. Therefore, the estimates are the mean increase in BMI per 1 unit increase of the metabolite.

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
