# Peer review of "A Workflow for Missing Values Imputation of Untargeted Metabolomics Data"

_metabolites, 2020, doi:10.3390/metabo10120486_

Round 1

Reviewer 1 Report

# Review of Faquih et al.

A Workflow for Missing Values Imputation of Untargeted Metabolomics Data

## Summary

In this manuscript, Faquih and co-workers attempt to extend the work previously performed by Do et al. on evaluating and applying two imputation methods, KNN-obs-sel and MICE-pmm on metabolomics data (more specifically, Metabolon data). They do so using the NEO dataset and simulations of missingness based on this dataset. Key advances include: determination between xenobiotic and endogenous metabolites should have different imputations; availability of an R-script for others to use.

I appreciate the authors' thoughtfulness in this manuscript and the work involved. I also appreciate that the authors took the time to create an R script that could be used by others.

## Major Issues

1. None of the supplementary materials, namely "Appendix A", nor Tables B1-B5 and Figures B1 and B2 are present.

2. No data is provided. I realize that the NEO study metabolomics data may not be generally available or is proprietary (which is likely based on my own searching for the data), but fake data with similar characteristics could have been generated based on the NEO data, and then used for simulation reporting at least. In addition, data could have been provided as example data for how to run the provided R scripts. Literally right now, I don't know what the data input to the script functions should look like, because I don't have Metabolon data .... I can guess, because I can read code, but lots of people can't.

3. Code for the simulations is not provided. Hand in hand with no data, the code used to actually create simulations and calculate the performance measures to evaluate the different levels of data missingness and size is not provided.

4. Figure 1 does not help the reader understand the distribution of missingness on a per class of metabolite level. I think this is primarily due to the differences in the number of metabolites of each class. The distributions in Figure 2 are much more helpful, and direct comparison would be facilitated by scaling to the number of metabolites in that class, and having each one directly underneath the other.

5. What information does Table 2 and Figure 3 actually provide? If the authors think the values of the mean correlation with 10 auxiliary metabolites are informative, then plotting the distributions might be more useful. I would recommend trying a "sina" plot with each category, or plotting the actual distributions of mean values instead of binning them.

6. Why the selection of 10 closest auxiliary metabolites? No justification for that number is provided, and no simulations used different numbers, even though the R script functions could be modified to have that as a variable.

7. Why did the authors choose "mean absolute correlation"? Is there an expectation that negatively correlated metabolites provide information about the value of the metabolite with missingness? Or does that only apply after rescaling to mean of 0 and standard deviation of 1?

8. Related to Point 5, if most metabolites have such low levels of inter-correlation with their auxiliary metabolites, then why does the imputation method work in spite of them? i.e. should we expect poor performance of the imputation methods with such low correlations?

9. Selection of metabolites: Would the authors have had more options if they had not concentrated on an association with BMI? From the description of the "mice" method, it is able to use covariates to improve predictions, such as BMI. There is an option in the R script to **not** use covariates, but the manuscript doesn't describe such a scenario. Why did the authors not test another set of metabolites with no association with BMI and turn off the option in their simulation?

10. Based on my own understanding, the kNN method uses no covariate information, but mice does. Given that the authors selected metabolites with strong association with BMI, is the comparison of methods fair? Especially because it seems in almost all the plots kNN did worse ....

11. Selection of subset datasets: The authors state they used a subset of 50 and subset of 150 samples to evaluate performance under different numbers of samples. In the selection of samples for a subsetting, was any consideration of matching the distribution of BMI / sex to the original distributions given? If not, should it have been? Is there any reason to expect differing performance with different subsets of samples? The authors don't state it, but I assume only 2 subsets (one of 50 and one of 150) were generated for testing? Does the set of 10 auxiliary metabolites change in the subsets compared to the full dataset?

12. It is not 100% clear that the base regression coefficients are from the subset datasets. Based on the description, I think this is the case, but the language could perhaps be clearer. My initial impressions were that only the regression coefficients from 599 set were used for comparison.

13. Tables 3 and 4: Should the **known** regression coefficients from each complete dataset be provided for comparison? Also, I cannot follow how Tables 3 and 4 tell me anything about how well the methods did, and how that follows from the text in the Results. Maybe I'm missing something in the explanation of what is in the tables or what is in the Results text that discusses them?

14. Is 5% percentage cutoff a good cutoff? Although I've not read a ton of imputation literature, I've read some, and this is the first time I've seen number of simulation cases where median value <= 5% were counted. Also, the total number of cases Y (where we have X/Y) seems to change in the results, but that doesn't make sense given there are the same number of simulations in each method of creating missing data. I'll admit, I'm totally baffled by the counts of less than 5%, especially without the error bars in the plots. Ideally those wouldn't be points, they would be boxplots, or sina plots with all the data points showing the distribution.

15. I feel like Figures 4 and 5 are attempting to do too much. They are attempting to show trends with respect to sample size, % missingness, method, and method of simulating missing values. While they do show overall trends, have the authors considered breaking out specific subsets to directly show these relationships in the supplemental? Or maybe that's what is present in the supplemental figures and tables that I can't find in this submission? Connecting the statements in the results sections to Figures 4 and 5 is currently very difficult personally, which also makes it hard to double check the claims made and their correctness based on the plots available.

## Minor Issues

1. The GitHub repository for the R script is buried in the references. In fact, the reference points to the Zenodo archive, not the GitHub repository. I'm glad to see an archived version of the script on Zenodo, which I'm assuming is the version used for the manuscript. However, others will benefit by having both the GitHub address directly in the abstract and at the end of the Methods, and then reference the Zenodo archive DOI as the version used for the manuscript results.

3. Pg 6, Line 158: This paragraph does not feel like it fits in this section. Perhaps another *Results* section titled "Availability"?

4. Pg 6, Line 163: I think this probably belongs as the first paragraph under *Evaluation*. To belong under *Characteristics*, it should be stated differently.
"12 metabolites (3 endogenous, 9 unnanotated) in the dataset had > 90% missingness and were subsequently treated as xenobiotic and imputed to 0".

5. Page 15, Line 397: Percentage bias is defined twice in two different ways. I'm sure only one was used given the plots provided. Perhaps one is supposed to be "Percentage RMSE"??

6. Was 1000 simulations for each necessary? Could 200 been done, and then 5 different random subsets of samples of each size (or more fine grained sizes) been done (see Point 5 of Major Issues).

7. Given the number of simulations, why are there no error bars in figures 4 and 5?

8. The color schemes in Figure 4 and Figure 5 are different. Given the methods are the same, why aren't the colors the same?

9. Why no 5% bias cutoff shown in Figure 5 when it seems like those numbers are discussed in the Results? Relatedly, would a simple horizontal line work better than the shaded area in Figure 4?

## Spelling

1. Pg 13, Line 346: "For non-xenobiotic metabolites, we assumed that they are metabolites with truly missing values
only if 90% of values were missing." I think the authors missed "only if less than 90% of values were missing".

2. Pg 14, Line 358: package "MICE" should be "mice" and package "vim" should be "VIM".

3. Pg 14, Line 380: "peforamance" should be "performance"

4. Pg 15, Line 397: "by the true estimate 27 ; The RMSE is the", "The" should not be capitalized

## Other Issues

The code provided is an R script and not an R package. I understand that R packaging can be a barrier, but it would ideally lead to more use, even if only hosted on GitHub. 98% of the work is already done given the level of documentation provided using Roxygen commenting! NOTE: I don't think the authors need to create an R package before this article can be published, but I would heavily encourage the authors to think about an R package instead of just a script. As it stands, this script could heavily wreak havoc with a users workspace in ways that most R programmers do not expect.

Reviewer 2 Report

The authors in this study provided an imputation workflow and a user-friendly R script to streamline the imputation of the Metabolon™ HD4 panel using kNN-obs-sel and MICE-pmm. The authors also evaluated the imputations using measured metabolomics data from the Netherlands Epidemiology of Obesity (NEO) study.

The study is interesting and offers novel insights for the metabolomics data users. I have the following concerns on this study;

  1. The authors did not address what causes the variation in the missing values in endogenous metabolites, xenobiotics and un-annotated metabolites.
  2. What is the significance of pattern in missing values in endogenous metabolites, xenobiotics and un-annotated metabolites? 
  3. In general, metabolomics analysis from normal individuals also produces xenobiotic metabolites. There are approximately 49% missing values in the xenobiotic metabolites. How relevant are these metabolites considering that the samples are from the obese and normal individuals who are not under any medications? 
  4. The univariate and multivariate imputation method for endogenous and xenobiotics metabolites can be included in the abstract.

Round 2

Reviewer 1 Report

I thank the authors for addressing all of my concerns, especially around the ability for others to be able to run code and see what is output. I believe the manuscript is much, much improved.

Reviewer 2 Report

The authors answered all the questions raised during the first review. Hence, the manuscript can be accepted.